# Sensors in the Autoclave-Modelling and Implementation of the IoT Steam Sterilization Procedure Counter

**DOI:** 10.3390/s21020510

**Published:** 2021-01-13

**Authors:** Lukas Boehler, Mateusz Daniol, Ryszard Sroka, Dominik Osinski, Anton Keller

**Affiliations:** 1Faculty of Electrical Engineering, Automatics, Computer Science and Biomedical Engineering, AGH University of Science and Technology, PL-30-059 Krakow, Poland; daniol@agh.edu.pl (M.D.); ryszard.sroka@agh.edu.pl (R.S.); 2Department of Electronic Systems, Norwegian University of Science and Technology, NO-7491 Trondheim, Norway; dominik.osinski@ntnu.no; 3B.Braun Aesculap AG, D-78532 Tuttlingen, Germany; anton.keller@aesculap.de

**Keywords:** sterilization, Internet of Things, sensors, simulation, internet of medical things

## Abstract

Surgical procedures involve major risks, as pathogens can enter the body unhindered. To prevent this, most surgical instruments and implants are sterilized. However, ensuring that this process is carried out safely and according to the normative requirements is not a trivial task. This study aims to develop a sensor system that can automatically detect successful steam sterilization on the basis of the measured temperature profiles. This can be achieved only when the relationship between the temperature on the surface of the tool and the temperature at the measurement point inside the tool is known. To find this relationship, the thermodynamic model of the system has been developed. Simulated results of thermal simulations were compared with the acquired temperature profiles to verify the correctness of the model. Simulated temperature profiles are in accordance with the measured temperature profiles, thus the developed model can be used in the process of further development of the system as well as for the development of algorithms for automated evaluation of the sterilization process. Although the developed sensor system proved that the detection of sterilization cycles can be automated, further studies that address the possibility of optimization of the system in terms of geometrical dimensions, used materials, and processing algorithms will be of significant importance for the potential commercialization of the presented solution.

## 1. Introduction

The Internet of Things (IoT) is an emerging technology that manifests a formidable potential to increase the safety, productivity and comfort of the future society as well as to disruptively change the future industrial landscape. However, the challenges for introducing this branch of technology vary from each field of application. While in consumer applications as much information as possible shall be available to increase usability and flexibility, the focus in healthcare and industrial applications relies preliminarily on reliability and low costs. Furthermore, in the medical field, sterile realizations of IoT systems need to fulfil the most challenging requirements due to the need of multiple steam sterilization cycles using temperatures up to 140 ${}^{\circ }$C and rapidly changing pressures. Due to these extraordinary requirements, most developments and research mainly focus on applications outside the sterile area of an operating room. Examples are the possible use of sensors, transponders, and smartphones to enable a multitude of different applications like tracking and utility management as investigated by Gaynor and Waterman [1]. Further research focused on the use of passive Radio Frequency Identification (RFID) to avoid the need of power sources. This enables maintenance-free identification and tracking applications within a hospital, as shown by Liu and Gu [2].

### 1.1. Current Issues of the Sterilization Process

While considering the challenges caused by increasing numbers of cases and growing stocks of medical instruments, a focus must also be put on patient safety during surgery. The use of reusable instruments creates the risk of failure during sterilization, which in the worst case can result in a serious infection. Studies have shown that successful sterilization depends on many different parameters. Both the selection of the process like in Winter et al. [3] and the preparation shown by Veiga-Malta and Rauwers et al. [4,5] are crucial for the elimination of all microorganisms. If required temperatures or holding times are not met, Panta et al. [6] showed that it will increase the risk of infection. For the monitoring and validation of steam sterilization processes, various indicators are utilized in the state-of-the-art systems. One example is using biological indicators which include microorganisms that change a measurable resistance during sterilization through their decease. However, this resistance change is not a reliable method for demonstrating compliance with maximum temperatures and their holding times. Another method uses chemical indicators signalling via colour changes to show whether a temperature limit has been exceeded. However, even this method of detection does not provide a reliable indication of the actual conditions. Physical methods that can be divided into active and passive indicators seem to be an easier solution. Sensors can measure parameters such as pressure, temperature, and humidity with high accuracy, but are also difficult to shield from the influence of high temperatures, as illustrated by Childers et al. and Pagan et al. [7,8]. The so-called Bowie and Dick test passively detects evacuation and steam saturation, which is a good indication of successful sterilization. However, incorrect results can also occur here due to altered temperature and pressure ramps and lead to a false positive test result, as it was investigated by the works of Laranjeira et al. and McCormick et al. [9,10].

### 1.2. State of the Art for Sterilization Process Validation

Approaches for reusable sterilization indicators mostly focus on mechanical solutions. Hereby, bimetals are used to detect a specified temperature. The disadvantage of this technology is the varying change temperature of the metal, causing low reliability for this solution. A further approach by Schuster and Schulz et al. [11,12] is the use of pressure-activated switches. However, detecting only pressure limits is not a reliable method to ensure successful sterilization. The reason for the shortage of reusable electronic systems is the lack of reliable thermal insulation. State of the art is the use of epoxy resin or high-temperature silicone. However, repeatedly sterilizing these materials increases the risk of an early failure of the electronic module due to damages in the insulation layer. Research showed that, during the encapsulation process, air encapsulations cause severe damage during the sterilization—in particular, using a nonflexible epoxy resin that requires a complex setup. The issue, in this case, is that failures often occur after a multitude of sterilizations, causing unexpected damage in the system as shown in previous works by Böhler et al. and George and Barrett [13,14]. In addition to the protection of the electronics from high temperatures, a long runtime has to be ensured. For use in clinical applications, a sensor system must either function for a very long time using a primary cell or, as was investigated in earlier studies by Daniol et al. and Lech [15,16], be designed to be autonomous with the help of thermal energy harvesting. Other approaches, for example, made by Rajasekaran et al. [17], consider automated wireless charging of sensors, but this creates further challenges that should not be considered in a first step.

Miller et al. [18] show an approach for a removable sensor module to ease the exchange and readout. Another focus is on the type of data transmission. Due to the need for thermal insulation and application area, wireless transmission is the preferred standard in most systems. In this case, the data obtained can be stored as a whole and transmitted after the sterilization is completed so that further evaluation can be performed, as shown by Hung et al. [19]. An overview is shown in Table 1.

### 1.3. Aim of the Work

To face the challenge of a reliable sterilization validation, a cost-effective sterilization counter shall be developed. This can improve safety and efficiency as well in healthcare as in industry. To save on costs and reduce the number of surgical sets, hospitals tend to rent sets from companies, paying just the amount of uses, which were performed. To ensure the correct amount of cycles, companies need to rely on the data given by the medical facility. By implementing independent counters, costs can be shrunk and the maintenance can be planned better to avoid a lack of medical equipment. Another essential aspect is defined by the Medical Device Regulation of the European Union. Here, for example, Article 22 (4)specifies that a sterilization process not specified by the manufacturer leads to a new conformity evaluation [20]. By ensuring that the specified sterilization parameters are adhered to, both the manufacturer and the applicator can be equally secured. Another key point is the lifetime of the sterilization counter. A primary cell was chosen to develop the prototype with the goal of a 30-day runtime. A crucial factor is hereby the choice of the communication interface. Active technologies like Bluetooth Low Energy bring the advantage to send over a long distance, even through covers; however, the power consumption of these technologies extends the capacity of a small profile coin cell. Passive solutions like RFID or Near Field Communication (NFC) have a very limited transmission range, but, due to the use of the reader’s energy, a very long battery life can be achieved. For the use in a simple counter, the data can be read out by close contact, so a passive technology is suitable. To be able to calculate the effectiveness of the thermal insulation, the developed system is tested in simulations. To ensure that these results can also be applied to subsequent developments, the simulation results must be compared with real measurements, like in previous studies of Daniol et al. [21]. This work is intended to create a platform that can be helpful in the design process of the next generations of the system.

## 2. Materials and Methods

To ensure reliable and autonomous detection of sterilization cycles, it is of utmost importance to ensure the resistance of the electronics for repeated steam sterilizations, which are both thermally and corrosively demanding on the sensor system. Despite the isolation, the system must be able to generate reliable sensor values to detect successful sterilization cycles. For the selection of the hardware, a deep analysis of the entire process has to be performed. For this purpose, temperature sensors were used in advance for the validation of autoclaves. Measuring the temperature inside the autoclave is sufficient for the first step because it can be used as a very good indicator for successful sterilization. There is no standardized sterilization process that would allow a uniform measurement for all processes. However, the minimum requirements for the sterilization process are specified. These requirements can be found in the relevant standards DIN EN 285, DIN EN ISO 17665-1, other specialized standards and in official information sheets. Important conditions here are a combination of the maximum temperature and its holding time. While at the temperature of 121 ${}^{\circ }C$, the minimum necessary sterilization time is 15 min, at 135 ${}^{\circ }C$, this time is reduced to 3 min [22,23]. Furthermore, the use of steam sterilization is recommended here, as it achieves very good results due to its thermal properties [24,25]. For this work, standard sterilization was performed, which is used in most medical facilities. The result of the performed temperature measurement is shown in Figure 1.

In this process, the maximum temperature is 135 ${}^{\circ }C$, which is maintained for 5 min. The presented process can be divided into several distinct stages. After introducing the objects to be sterilized, the chamber is periodically filled with steam and evacuated. The temperature in the chamber reaches approximately 110 ${}^{\circ }C$. Once the chamber has been successfully evacuated, meaning that there are no more air pockets in the autoclave, the steam is heated to the maximum temperature. For this purpose, the pressure in the chamber is also increased to up to 3 bar. Once this is reached, a plateau time of at least 3 min is maintained. In most cases, this time is significantly extended to compensate for temperature fluctuations and to ensure that the required temperature is reached in areas that are difficult to heat. After this phase, the pressure is reduced and the chamber is cooled. Air continues to be introduced into the chamber until the objects can be removed. Understanding the typical sterilization process is necessary for the development of the sensor system that will confirm that the essential temperature plateaus can be detected and evaluated with little computational effort.

### 2.1. Prototype Design

When developing a design for the prototype, it was important to ensure contactless communication and a low energy consumption. The first step was the electronic design, followed by hardware design and software development.

#### 2.1.1. Electronic Design

For the selection of the electronic components, the vital factors are temperature resistance and the lowest possible power consumption. Since the analysis procedure is limited to the temperature acquisition and holding time, no complex arithmetic operations are required, thus it is feasible to select an ultra-low-power microcontroller. A wireless interface is also required for contactless transmission of data and software updates. Due to the short transmission distance and minimization of power consumption, passive NFC is preferred over active Bluetooth.

These requirements are met by the selected RF430FRL152H microcontroller (Texas Instruments, Dallas, USA). However, since its maximum operating temperature is only rated at 70 ${}^{\circ }C$, thermal insulation must be used. To ensure that the power can be supplied over a longer period of time, a $\mathrm{Li}/{\mathrm{MnO}}_{2}$ high-temperature primary coin cell is used, which is designed for up to 125 ${}^{\circ }C$. Since the battery voltage exceeds the permissible maximum tolerable input voltage of the microcontroller, a step-down converter with very low power dissipation and an operating temperature of up to 150 ${}^{\circ }C$ was used. The circuit board was constructed according to the technical manufacturer’s specifications using high-temperature components. Test points were also inserted for initial programming, which could be connected to a programming adapter. For the initial prototype phase, the NFC antenna designed by Texas Instruments was used, which mainly influenced the size of the printed circuit board (PCB), as shown in Figure 2.

#### 2.1.2. Mechanical Design

After completion of the electronic design, the mechanical design stage could begin. Since the sensor system had to be tested in several steps, a possibility of subsequent programming and a hardware change had to be ensured. For this reason, the choice fell on a screwed housing, which was equipped with an additional sealing ring to protect against penetrating liquid. The first version of the prototype has a relatively large housing due to the usage of a secure screw connection. However, in later versions, the sensor system is to be encapsulated, which will minimize the installation space and increase the robustness of the system.

The readability of the NFC interface is a requirement that has to be taken into account in the design process. The maximum layer thickness was determined experimentally by adding different spacers between antenna and reader. Hereby, 15 mm were found to be a safe transmission range. Furthermore, a suitable material had to be selected for the prototype. Due to the thermal requirements and the need for a multi-part housing, polyether ketone (PEEK) was selected. This thermoplastic is already used in medical technology and is ideally suited for the sensor system due to its low thermal conductivity of approximately 0.2
W/mK. The overall dimensions and buildup are shown in Figure 3. The prototype’s outer case consists of an upper (A) and a lower (D) round case made out of PEEK and eight steel screws (B) to secure the silicone sealing ring (C). Inside the case, the PCB with the electronic components (F) is protected downwards by a layer of aerogel-based thermal insulation (G). In the upper part, the high-temperature coin cell with holder (E) is placed.

#### 2.1.3. Preliminary Software Design

For the lowest possible energy consumption, the microcontroller must operate at minimum computing power. Furthermore, the FRAM memory with 2 KB requires an efficient data handling. For a first evaluation of the sensor output, a simple acquisition and storage of the ambient temperature was programmed. Acquired output values allowed preliminary determination of the offset and accuracy of the system by comparison with data from an external temperature sensor. The final implementation of the software for the detection of sterilization cycles was done after the detailed analysis of the results and simulations.

### 2.2. Thermal Modelling

To perform thermal simulations, a 3D model has been created. The CAD model has been simplified, and nonessential parts from the heat transfer perspective have been deleted. The model is presented in Figure 4.

It consists of PEEK housing connected with stainless steel screws with silicone rubber sealing in between. Inside the prototype, there is a PCB with a sensor and antennas on one side and the coin cell battery on another side. The main difference in comparison to the real-world prototype is that all SMD resistors and capacitors have been removed. PCB and coin cell are surrounded by aerogel-foam-based insulation. To compensate for the lack of SMD parts, the PCB thermal properties have been adjusted according to the electrical design of PCB. Two main materials of PCB have been taken under consideration-PCB laminate and copper. The given PCB design consisted volumetrically of around 0.6% of copper and 99.4% of PCB laminate. Due to their high thermal conductivity in comparison to the housing material, stainless steel mounting screws have also been modelled. Simulations have been performed using the SimScale cloud simulation platform. All calculations have been performed using transient heat transfer analysis.

#### 2.2.1. Material Parameters Selection

Thermal parameters of the model components have been carefully selected and adjusted. In most cases, material tabular data have been chosen. However, in some cases, these have been not available as a single, precise value. Therefore, for the PCB and casing material (PEEK), the average values from the table ranges are assumed. In addition, the values of PCB parameters have been tuned due to the copper pathways on the surface. All of the material parameters are presented in Table 2.

#### 2.2.2. Battery Modelling

When modelling the thermal behaviour of the battery, not only the material properties of the object but also the electrochemical properties must be taken into account. In the described model, the authors used a simplified diagram of the cell. It was caused by the lack of factory data and construction diagrams. The simulated cell consists of a metal casing, a layer of a lithium anode, a separator, and a layer of manganese dioxide cathode. A plastic seal is placed on the connection of the housing. Due to a very low power consumption from the battery, the effect of self-heating during the operation was omitted in the analysis, significantly simplifying the simulation problem. The selection of thermal parameters of each layer of the battery had to be made using the average of the array values. It was assumed that the casing is made of 304-grade stainless steel. It was further assumed that the material constituting the anode is lithium with its thermal parameters. A separator in the cell separates the anode from the cathode. It is usually a polymer-based material with strong thermal insulation properties. In the described model, it is an important factor limiting the heat flow on the vertical axis of the cell. Based on a literature review of the separators, it can be seen that their thermal parameters depend to a large extent on whether they are dry or wet. The thermal conductivity of the separators takes the value from 0.07 to 1.45 W/mK [26,27]; for the purpose of the conducted simulations, the value of 0.5 W/mK has been selected. In the works [28,29,30], the authors estimate the specific heat of the separators in the range from 1838 to 1987 J/kgK; in the work [31], the value of 700 J/kgK has been assumed and in the work [27] 2438 J/kgK. For the needs of this simulation, 2000 J/kgK has been chosen. The density of materials used as a separator is in the range 913–1200 kg/$m^{3}$ [27]; for the simulation, the value of 1000 has been chosen. In case of the cathode, most of the thermal parameters have been estimated, as, despite authors’ best attempts, it was not possible to find relevant research on the Electrolytic Manganese Dioxide (EMD) used in this battery type. According to the manufacturer’s safety data sheet [32], the cathode is composed of Manganese Dioxide (${\mathrm{MnO}}_{2}$) soaked with Propylene Carbonate (PC—$C_{4}H_{6}O_{3}$), so the thermal parameters of the ${\mathrm{MnO}}_{2}$ cathode will be affected by the electrolyte. Based on the research of Richter et al. [33], it can be assumed that the thermal conductivity of the soaked cathode might be even 2–3 times higher. Hedden et al. [34] estimates the thermal conductivity of $fi-{\mathrm{MnO}}_{2}$ nano phase as 4.0
W/mK. However, the whole cell contains 2–9% of PC with thermal conductivity of 0.14
W/mK [35], which will affect the thermal conductivity of ${\mathrm{MnO}}_{2}$ cathode. Thus, the authors took $\kappa_{MnO_{2}}=\;$
3.5
W/mK for the purpose of thermal simulations. Specific heat and density of ${\mathrm{MnO}}_{2}$ have been taken from [36,37], and was set to 594 J/kgK and 5026 kg/$m^{3}$, respectively. Finally, heat can be generated by the cell itself, while discharging it with a current exceeding its maximum discharge current limit. For CR2450HT (RENATA, Itingen, Switzerland), these limits are 0.8 and 3.0 mA for continuous and maximum discharge current, respectively. The authors expected a very low current consumption of the system, within the standard discharge current limit of the cell. Thus, to simplify simulations, the heat generated by electrochemical phenomena during battery discharge was neglected. All of the battery material parameters chosen for simulation are listed in Table 2. Cross section of modelled battery is presented in the Figure 5.

#### 2.2.3. Simulation Parameters

The subject of the simulation was the temperature distribution and heat flux inside the model. All simulations were conducted on a SimScale cloud platform. For numerical calculations, FEA Code Aster was used. It is a solver specialized in thermomechanics and heat transfer in solids. For the simulation, a properly prepared, simplified CAD model with finite volume meshing method was used. Initially, the global temperature of the model was 101 ${}^{\circ }C$. It was consistent with the measurements made on the prototype heated in the thermal chamber. As the subject of the simulation was the temperature distribution inside the model, the temperature of the prototype walls during a given unit stroke was set as boundary conditions. These conditions were set for all external surfaces of the model. The simulation used a MUMPS solver with the SCOTCH renumbering method and a Theta parameter of 0.57 due to the relatively complex geometry of the model. The simulated process took 4000 s and the time step length was 15 s. During the simulation, the temperature was measured at several control points, including the housing, battery, and thermistor location for quick prevalidation. Full simulation data were saved in a VTM multi-block file and then processed in ParaView.

### 2.3. Simulation Validation

The results of the simulation were validated in the laboratory by applying a thermal unit step to the prototype and recording its response. The prototype was heated in a thermal chamber to a temperature $T_{\mathrm{hot}}=$ 100 ${}^{\circ }C$ and then taken out for a cooling station to the laboratory temperature $T_{\mathrm{amb}}=$ 27 ${}^{\circ }C$. Two thermocouples were attached to the model—one to the housing and the other to the thermistor. The thermocouples were connected to the temperature logger TC-08 (Pico Technology Ltd., St. Neots, UK). The whole procedure is presented in Figure 6. The measurement error of the device was calculated based on the formula provided by the manufacturer:(1)$$E_{TC08}=0.2\frac{T_{hot}-T_{amb}}{100}+0.5^{\;\circ }C$$

The thermocouples connected to the device, following the IEC584-2 standard, have an accuracy of $E_{\mathrm{KT}}=\;$ 1.5 ${}^{\circ }C$. The overall measurement error of the system can therefore be described as:(2)$$E_{MS}=\sqrt{E_{TC08}^{2}+E_{KT}^{2}}=1.28^{\;\circ }C$$

The measurements from both thermocouples were recorded at a frequency of 1 Hz using PicoLog 6.1.17 software. The measurement lasted until the thermal state of the prototype was stabilised, which occurred after 73 min. Then, the time scale of simulation and measurement was unified. The data collected from the thermocouples were subsampled using the last sample method (function available in the PicoLog software). Finally, the measurement results were compared with the simulation results, taking into account a possible measurement system error. The results of this validation were used to assess the quality of the identification of the simulated model.

#### 2.3.1. Thermistor Calibration

The RF430 microcontroller has very limited calculation possibilities. Reading from an analog-to-digital converter and converting this value into degrees Celsius is not possible in the device memory. Therefore, it was necessary to prepare a table of values corresponding to the individual temperature values. To create this lookup table (LUT), it was necessary to collect the measurement of the raw ADC values and then to carry out the conversion using a calibration formula in the sensor’s operating range (20 ${}^{\circ }C$ to 140 ${}^{\circ }C$). In the implemented solution, which is presented in Figure 7, a thermistor with R = 100 kΩ was connected to a 14-bit ADC with a reference resistor $R_{\mathrm{REF}}=\;$ 100 kΩ. During each calibration measurement, the value was read from the reference resistor and the thermistor. Based on the manufacturer’s datasheet [38], the following calibration formula was used to create the table:(3)$$T_{C}=\frac{1}{\frac{\frac{\log_{10}\frac{R_{conv}}{100000}}{\log2.718}}{4330}\;+\frac{1}{298.15}}-273$$
where
(4)$$R_{conv}=\frac{U_{TR}}{U_{REF}}\;\cdot \;R_{REF}$$
with $U_{\mathrm{REF}}$ being a voltage on reference resistor, $U_{\mathrm{TR}}$ being a voltage on the thermistor, and $R_{\mathrm{REF}}$ being the resistance of a reference resistor. To measure the temperature, the data from the ADC are compared each time with the table values and adjusted to the respective value range.

#### 2.3.2. Thermistor Accuracy

The measurement system used in the prototype has a measuring range from 5 ${}^{\circ }C$ to 70 ${}^{\circ }C$ due to the limitations of the measuring track construction. However, the expected temperatures inside the sensor exceed this range. Therefore, an experiment was carried out to test the sensor’s performance in the range up to 100 ${}^{\circ }C$. A thermocouple was attached to the sensor in the place where the thermistor is located. At the thermocouple-thermistor contact, a thermally conductive paste was used. The sensor was connected by a JTAG interface to the computer. Integrated Code Composer Studio development environment was used to preview read values during program operation. Values read from ADC were saved every second to a file with an appropriate timestamp using GEL script. Synchronously, Pico TC08 device readings from the thermocouple were saved in the PicoLog application. To determine the accuracy of the sensor, its thermal step response was measured. The sensor was heated in the thermal chamber to 100 ${}^{\circ }C$ and then removed to the cooling station recording readings from ADC and thermocouple. Finally, the readings from the thermocouple and ADC were compared, determining the limit point at which the ADC maintains the assumed accuracy.

#### 2.3.3. Software Design

Based on simulation results and measurements of the accuracy of the prototypes’ measurement system, appropriate software has been developed. It has been designed for basic recognition of the sterilization process and counting of their quantity. The algorithm reads the data from ADC and then compares them with the values stored in the LUT. Registered temperature is the last value from the table from which ADC was higher. If the read temperature exceeds the assumed threshold value, it is assumed that the sterilization procedure has started. Then, the values are written to the array in the FRAM. If the threshold has been exceeded in the other direction, it means the end of sterilization, and the counter is incremented and saved in the FRAM memory. This memory does not need the power to store data; therefore, between the login process and the reading procedure, the entire sensor can remain in a standby state. When reading with an RFID reader, the entire chip is excited and the data from the FRAM memory are read and sent as an NFC Data Exchange Format (NDEF). After the reading procedure is completed, the memory is cleaned and the sterilization counter is reset. A diagram describing the whole software workflow is depicted in Figure 8.

#### 2.3.4. Power Consumption

One of the requirements for the proof of concept was to operate the sensor for at least 30 days on one CR2450 battery. For this purpose, basic energy-saving algorithms were used. While the sensor is in standby mode, it reads the temperature at ADC every 10 min. If the temperature exceeds the 40 ${}^{\circ }C$ threshold, the reading interval changes to 1 min. If the 60 ${}^{\circ }C$ threshold is exceeded, it changes to 15 s. After completion of the sterilization cycle counting procedure, the device goes into standby mode. In this state, the device is most of the time in a deep sleep state (LPM4 mode of RF430 chip), waking up only during data acquisition from ADC. Due to the use of FRAM memory, the power supply is not required for data storage and its reading is possible using the energy recovered from the electromagnetic field emitted by the reader. This guarantees that, even when the battery is completely discharged, the data will not be damaged or lost.

### 2.4. Algorithm Validation

The developed algorithm was validated using simulated test data resulting from the simulation. The algorithm has been transcribed to the Python language while retaining its features relating to the energy-saving mechanism. The temperature curve of the sterilization process has been placed in the table. The board moved at the frequency of 4 times per minute. Simulated input of ADC converter was the value from under index 0 of the board. The algorithm worked according to its design by reading the input of simulated ADC at a specific frequency and processing the obtained values. Due to the early stage of development of the project, it was only checked if correct sterilization processes would be counted.

## 3. Results

The recorded response of the model sensor to the unit temperature jump is shown in Figure 9. As can be seen in most cases, the discrepancy between the simulations and the measurement collected from the prototype is within the limits of the measurement system acquisition error. A slight overrun is visible at the very beginning. It may be caused by inaccuracies in the selection of material parameters of the battery and PCB due to unavailable tabular data. In general, the difference between simulated and measured values does not exceed 2 °C at any time, which is considered sufficient for the application described in this paper.

Having a sensor model developed and verified, we have performed thermal step response measurements to see the comparison between the temperature acquired by the sensor software and the values from a thermocouple attached to the thermistor. As can be seen in Figure 10, dynamic measurements have shown that the values from the thermocouple coincide with the values read by the sensor software up to a temperature of about 70 ${}^{\circ }C$. Above this temperature, the measurements differ significantly. This is due to the design of the sensor’s analog front-end. The main limitation comes from the limits of the ADC rail (0.9V) which the 100k thermistor, with a 3μA bias current, starts to hit the high voltage limit rail at $70^{\;\circ }$C. This limit cannot be improved by much, such as going to below 0 ${}^{\;\circ }$C since the RF430FRL15xH recommended operating temperature range is 0–70 ${}^{\circ }C$. The measuring resolution depends on factors such as the oversampling time and the programmable gain amplifier settings. In general, according to the available documentation, the SD14 ADC using a thermistor can achieve accuracy of 0.5 ${}^{\circ }C$ in the 20–45 ${}^{\circ }C$ range without calibration.

Step response analysis showed that the identified model can be used for further simulations. Both material and simulation parameters have remained unchanged. As boundary conditions, the temperature on the surface of the walls of the prototype placed in the autoclave, measured previously with a calorimeter, was given as the temperature on which a standard sterilization procedure was performed. Figure 11 shows the course of simulated sterilization procedure and simulated temperature course on the thermistor.

As shown, the maximum temperature reached by the thermistor is 116 °C. A shift of the peak in relation to temperature is also visible on the prototype’s surface. It is caused by relatively high thermal inertia of the model itself, built mainly of PEEK. It is very important that the temperature on the thermistor placed in the immediate vicinity of the battery has not exceeded 125 °C, for which the CR2450HT battery is certified. The drawing also shows the limit temperature that the prototype’s measurement system can acquire. Based on the data depicted in Figure 11, a timeframe with the highest temperature gradient between the sensor region and casing has been identified (t = 225 [s]). To identify potential areas for optimisation, simulation data were visualised showing temperature distribution and heat flux. This made it possible to find probable thermal bridges. Elements that bring significant heat into the PEEK enclosure are stainless steel screws (a in Figure 12B). In addition, it can be seen that the aerogel layer insulating the battery from the top seems to be too thin—there is a relatively high heat transfer (b in Figure 12B) compared to the lower, two-millimetre Aerogel layer. Other heat bridges identified on the simulation data include the contact between the edges of the PCB and the casing (c in Figure 12B) and the battery holder made of PET (d in Figure 12B). The temperature distribution inside the prototype shows that the battery is relatively well insulated from the external environment (battery in Figure 12A). After the simulations and thermal analyses, the operation of the algorithm in the simulated test environment was verified. These tests confirmed the correct operation of the algorithm and the counting of sterilization cycles. Tests of the sensor’s power consumption made with the use of Code Composer Studio software (Texas Instruments) showed that the sensor consumes on average 0.6 mW of power during operation. It means that the operating time on a CR2450 battery, with a capacity of 490 mAh, will be 101 days. Considering the very limited range of power consumption optimizations made, and keeping on the CPU because of other debugging tasks ongoing, it should be assumed that this time may be extended.

## 4. Discussion and Conclusions

By comparing the simulated temperature values and the experimentally determined values, it has become clear that the selected simulation parameters are suitable for in-depth thermal analyses. The difference between simulated and measured temperature curves was consistently less than 2 ${}^{\circ }C$. Therefore, the simulation can be used to predict real measurement results and establish a platform for valid case testing.

The results of the simulation show that the thermal insulation of the prototype needs to be improved. In order to provide better protection for the electronics, the heat transfer must be reduced. Therefore, the current issues of the state of the art, described by Böhler et al. and George and Barrett [13,14], for reliable encapsulate electronics for the use in autoclaves need to be further investigated. The experimental results also show that the core temperature of the sensor system reaches 116 ${}^{\circ }C$. This temperature is particularly critical for the microcontroller. Its value exceeds the allowed operating temperature of the microcontroller, and, over the long term, could cause malfunction or even destruction of the device.

The detection of sterilization cycles by the developed software could be confirmed, but there are strong limitations due to the used hardware. Due to the limitation of the ADC evaluation caused by the used 100 kΩ thermistor, only a valid temperature measurement of up to 70 ${}^{\circ }C$ is possible. However, the results show that the approach of this study overcomes most of the current issues.

In addition to the temperature measurement, a power consumption analysis of the microcontroller was also carried out. Average power consumption of 0.6 mW was calculated. Using the CR2450 coin cell, a total runtime of over 100 days can therefore be predicted, which by far exceeds the required 30 days. This enables a reliable solution without the need of wireless charging or energy harvesting as mentioned by Daniol et al., Lech and Hung et al. [15,16,19] and enables therefore cost and size reduction.

The work presented here shows that an autonomous detection and counting of successful sterilization cycles can be performed by an electronic sensor system. The measurement results show that temperature peaks are detected; however, these do not fully correspond to the real values due to the limitations of the used hardware. For this reason, there is a potential for improvement in this area for further developments. Additionally, a revision of the thermal insulation is advisable in order to increase the lifetime of the electronic components.

The analysis of the selected hardware and developed software shows that a realistic application in a hospital is achievable. Both the significant exceeding of the targeted lifetime of 30 days and the possibility to design a reliable sensor system with minor improvements show that a cornerstone for valid detection of sterilization cycles was made by this work.

Important for further developments is the successful thermal simulation for the designed prototype. The selected parameters allow in-depth analyses of different designs and the detection of possible errors in sterilization processes. This enables the possibility of investigating the future development of the system as a device capable not only of counting the sterilization cycles but in addition able to verify normative thermotemporal criteria for the quality of the sterilization process.

Although due to the design constraints the current system is not able to detect rapid short-lasting temperature changes, the developed model for thermal simulations can aid the process of implementing necessary hardware changes for achieving a better thermal dynamic response of the system. The development of reliable detection of standard-compliant sterilization processes can significantly increase safety for patients, hospitals and manufacturers alike.

## Figures and Tables

**Figure 1 sensors-21-00510-f001:**
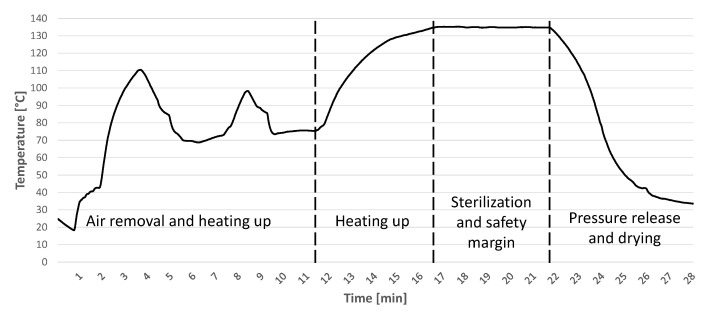
Temperature measurement of the standard sterilization process.

**Figure 2 sensors-21-00510-f002:**
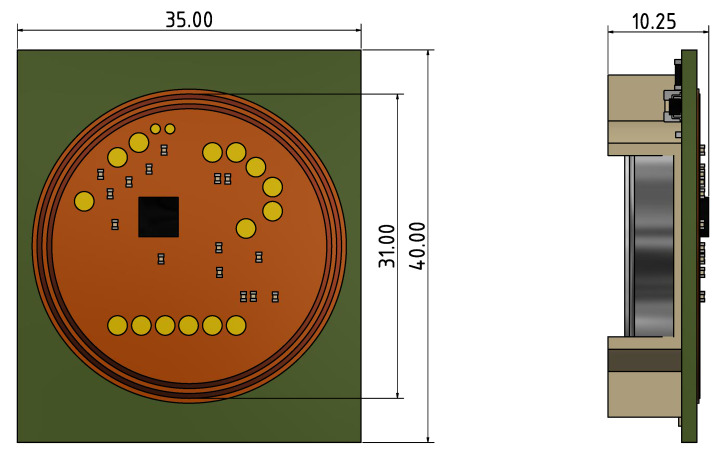
PCB cross section with dimensions.

**Figure 3 sensors-21-00510-f003:**
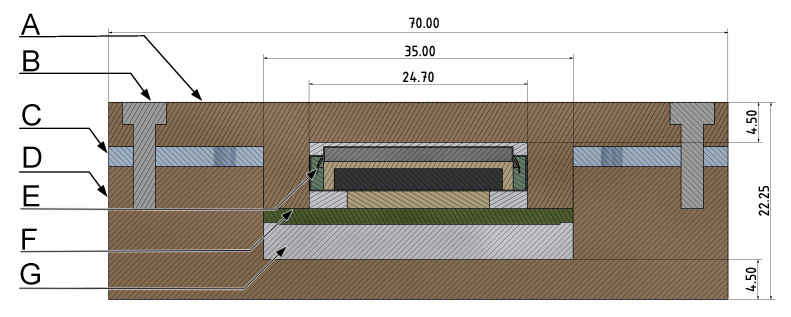
Cross section of the prototype with dimensions. (**A**) upper case, (**B**) steel screw, (**C**) silicone sealing ring, (**D**) lower case, (**E**) coin cell with holder, (**F**) PCB with electronics, (**G**) aerogel-based thermal insulation.

**Figure 4 sensors-21-00510-f004:**
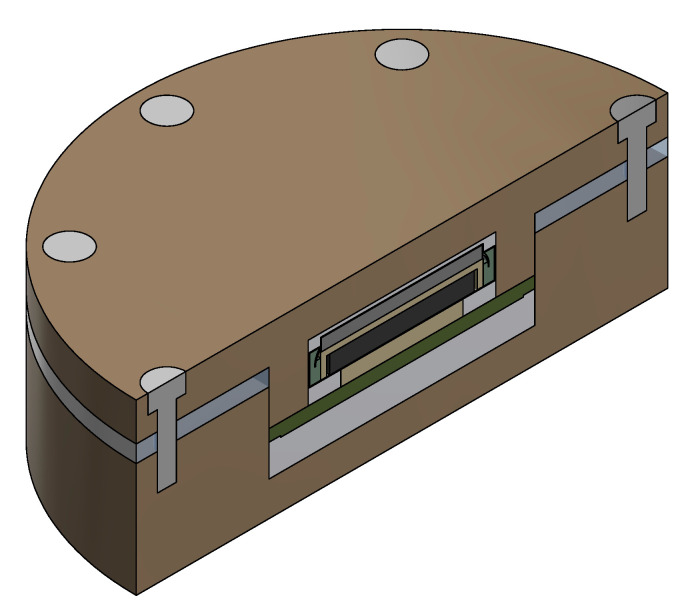
Simplified 3D model used for simulations.

**Figure 5 sensors-21-00510-f005:**
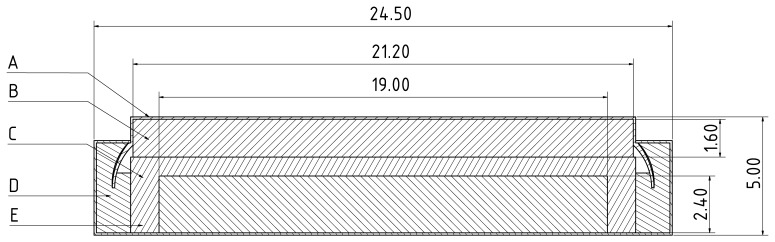
Cross-section through 3D coin cell model developed for simulation purposes. (**A**) steel casing, (**B**) Li anode, (**C**) separator, (**D**) gasket, (**E**) ${\mathrm{MnO}}_{2}$ cathode.

**Figure 6 sensors-21-00510-f006:**
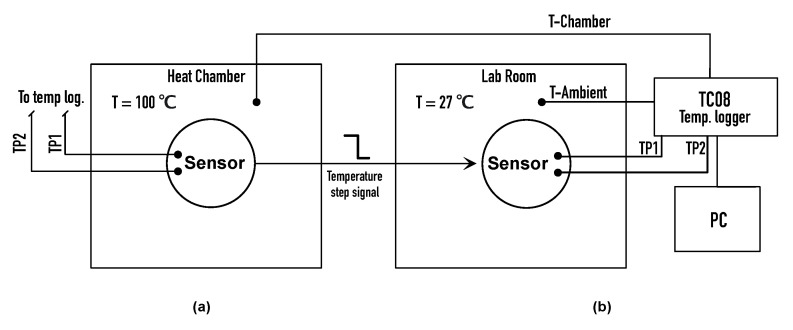
Measurement setup of a thermal step signal prototype response acquisition in thermal chamber (**a**) and in ambient temperature of laboratory room (**b**).

**Figure 7 sensors-21-00510-f007:**
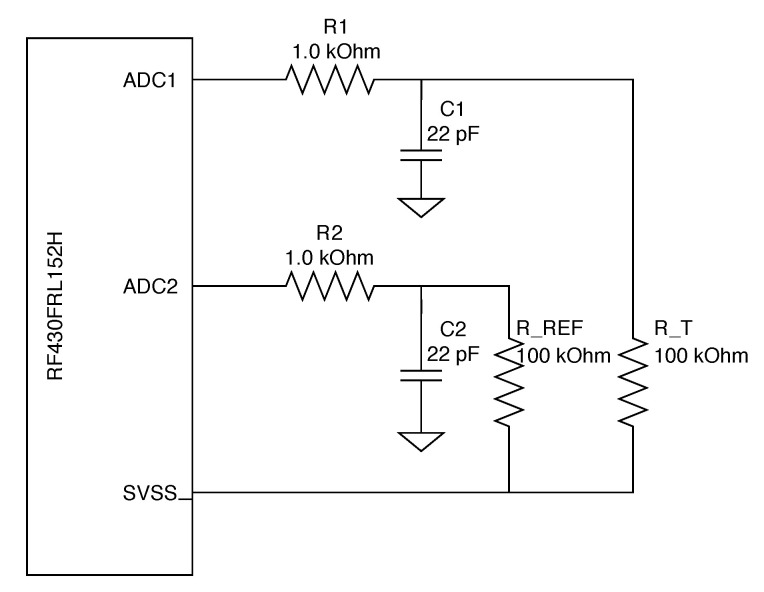
Schematics of the ADC temperature acquisition block with reference resistor R-REF and thermistor R-T.

**Figure 8 sensors-21-00510-f008:**
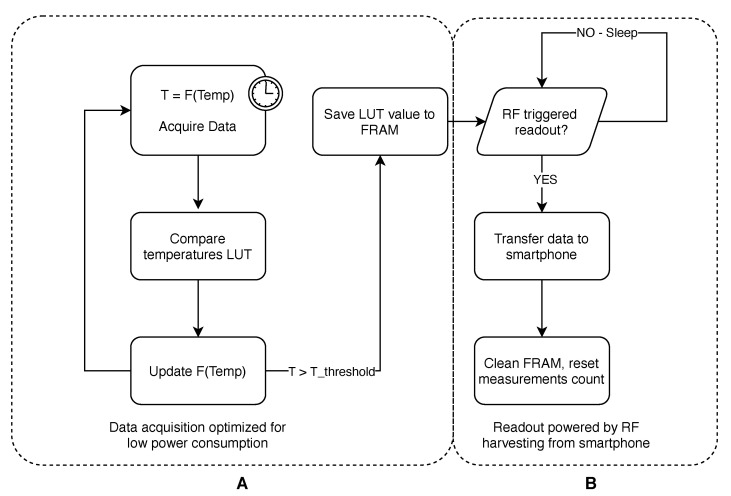
Overall diagram of the software. (**A**) data acquisition algorithm; (**B**) NFC readout procedure.

**Figure 9 sensors-21-00510-f009:**
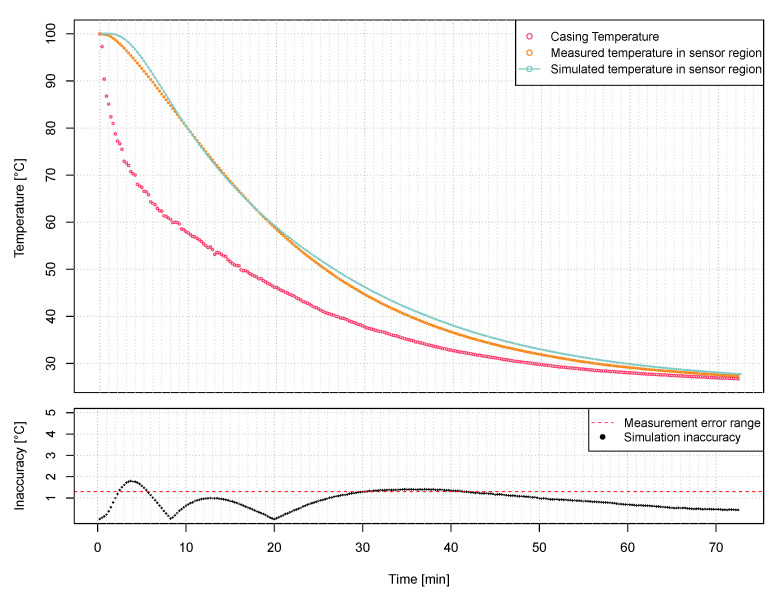
Results of thermal step response of the model and a prototype.

**Figure 10 sensors-21-00510-f010:**
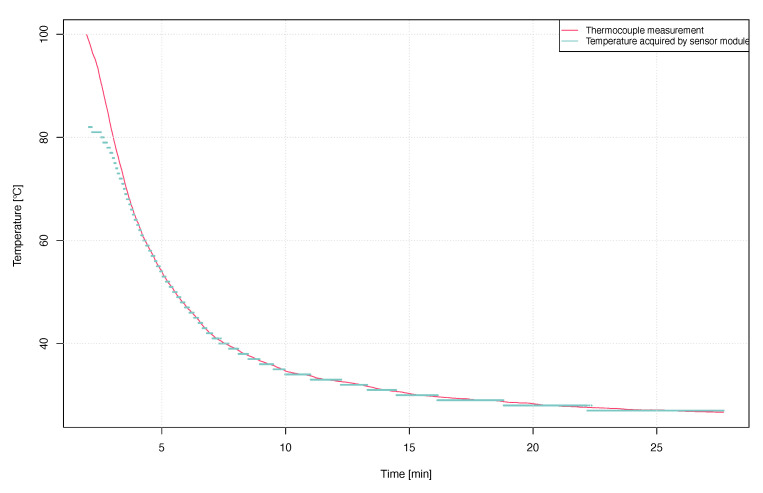
Thermal step response measurements comparison between thermistor with a custom measurement algorithm, and thermocouple results.

**Figure 11 sensors-21-00510-f011:**
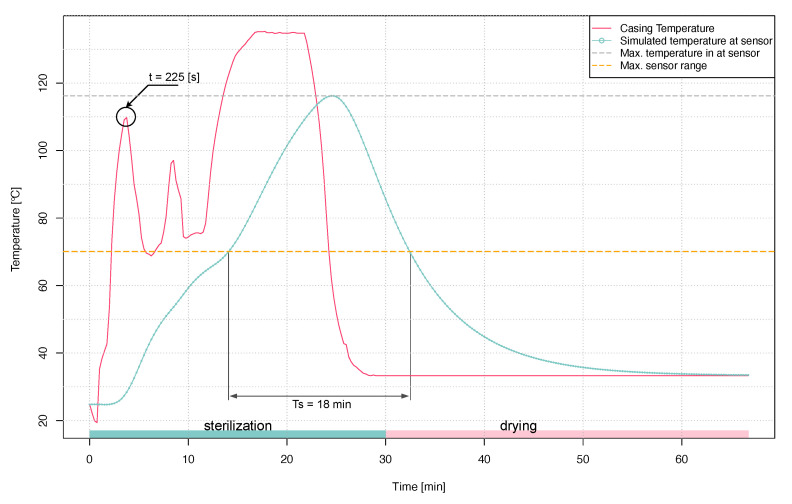
Simulated sterilization procedure and drying after sterilization.

**Figure 12 sensors-21-00510-f012:**
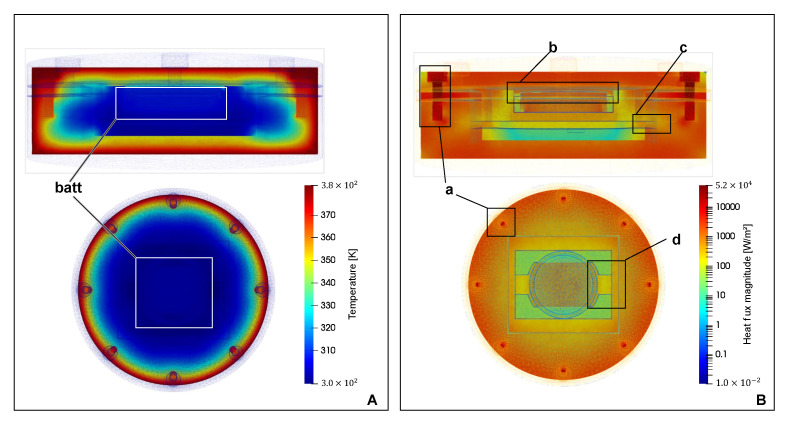
Simulation results—state after first temperature peak during the sterilization displaying temperature (**A**), and heatflux (**B**).

**Table 1 sensors-21-00510-t001:** Overview of the state of the art.

Study	Sensors	Power Supply	Communication	Insulation	Focus
Schuster	Mech. Pressure	none	Optical Indicator	none	Sterilization detection
Schulz et al.	Mech. Temperature	none	none	none	Sterilization detection
Böhler et al.	Temperature	Primary Battery	BLE	Epoxy Resin /Aerogel	Sterilization detection
George & Barrett	Temperature /Pressure	Primary Battery	433 MHz ISM	Epoxy Resin	Sterilization detection
Daniol et al.	none	Thermal Energy Harvesting	none	Aerogel	Power Management (sterile)
Lech	none	Thermal Energy Harvesting	Temperature	none	Sterilization detection
Rajasekaran et al.	Environmental Sensors	Wireless Charged Battery	RFID	none	Power Management (non-sterile)
Miller et al.	Environmental Sensors	Rechargeable Battery	Wireless	Plastic	Sterilization detection
Hung et al.	Temperature	no information	2.4 GHz ISM	none	Sterilization detection

**Table 2 sensors-21-00510-t002:** Material properties used for thermal simulations of developed model.

Material	Density/kg/m${}^{3}$	Thermal Conductivity/W/mK	Specific Heat/J/kgK
PEEK	1320	0.2	1400
Steel	8000	14.0	480
PCB	1700	0.25	1300
Silicone Rubber	1100	1.0	1300
Epoxy	1400	0.35	1000
PET	1100	0.15	1200
Aerogel	120	0.027	1500
Lithium	535	84.8	3582
Separator	1000	0.5	2000
MnO${}_{2}$	5026	3.5	594

## Data Availability

Data sharing not applicable.

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
