# Peer review of "Sensors in the Autoclave-Modelling and Implementation of the IoT Steam Sterilization Procedure Counter"

_sensors, 2021, doi:10.3390/s21020510_

Round 1

Reviewer 1 Report

This paper is related to IoT, and it is a subject that is currently researched. The affiliations should be corrected as the three first authors have the same affiliation. The introduction is very well-structured, but the state-of-the-art will be valued with a table to synthesize the results. Also, more recent related work must be presented, removing the oldest references. The methods and results are clearly presented, but they must be discussed with the literature.

These are the major flaws of the paper.

The paper is interesting, but the related work must be better structured with a diagram or a table. Also, the results must be discussed with the literature in related work.        

Author Response

Dear Sir or Madam,

First of all, we would like to express our gratitude to you for your explicit remarks

and constructive suggestions. We have carefully read your comments and made some

modification on the original manuscript according to your comments and suggestions.

We hope to be able to meet your expectations. We first quote your comments,

then give out our answers.

Our replies to you are as follows:

C1: The affiliations should be corrected as the three first authors have the same affiliation.

A1: The affiliations have been corrected according to the suggestion.

C2: The introduction is very well-structured, but the state-of-the-art will be valued with a table to synthesize the results.

A2: A table with the summary of the named state of the art was added.

C3: Also, more recent related work must be presented, removing the oldest references.

A3: The related work was investigated, and more recent works were added to the state of the art.

C4: The methods and results are clearly presented, but they must be discussed with the literature.

A4: The discussed literature was reflected in the discussion section.

Yours sincerely

Lukas Boehler,

Reviewer 2 Report

The work presented in this article attempts a solution for the development of a sensor system for automatic detection of steam sterilization based on temperature profiles. The solution is based on determining the relationship between the surface temperature of the tools and the temperature of the measurement points using a thermodynamic model. To validate the developed system, results of thermal simulations of the system have been compared with the measured temperature profiles, which show that automation of sterilization detection is achievable.

  • This is an interesting work considering the significance of the problem being addressed with regard to the sterilization of surgical instruments and implants, which involve human lives in terms of safety. The problem has been clearly defined, solution was proposed and fully described. The design and development of the solution have also been fully presented with validation results.
  • Another important part of the solution is the electronic design and hardware design of the system taking into account the low-power consumption and wireless communication of the system. Several important considerations have also been adequately explained.
  • Last but not least, another aspect of the work presented is the fact that the design and the implementation decisions ensure that energy-saving algorithms as well as components have been used. For example, the use of FRAM ensures that power is not required for data storage with data reading using energy recovered from the magnetic field emitted by the reader.

However, authors need to address the following concerns.

  • On line 186, in the  3D model of the thermal simulations, a simplified CAD model with nonessential parts have been removed. The resistors and capacitors have been removed in comparison to the real-world PCB. The authors thus claim that to compensate for the SMD parts, the PCB thermal properties have been adjusted. Provide rationale for this adjustment and explain what adjustments have been made.
  • Equation 3 is not clear, please re-write.
  • According to the authors, Figure 8 describes the whole software workflow/diagram of the software, however, the same figure refers to the data acquisition algorithm. Authors should clarify the difference.

  • On lines 159-150, authors claim that “The readability of the NFC interface is a requirement that has to be taken into account in the design process. The maximum layer of thickness was determined experimentally in previous work.” No reference to this work!

Author Response

Dear Sir or Madam,

First of all, we would like to express our gratitude to you for your explicit remarks

and constructive suggestions. We have carefully read your comments and made some

modification on the original manuscript according to your comments and suggestions.

We hope to be able to meet your expectations. We first quote your comments,

then give out our answers.

Our replies to you are as follows:

C1: […] The authors thus claim that to compensate for the SMD parts, the PCB thermal properties have been adjusted. Provide rationale for this adjustment and explain what adjustments have been made.

A1: An explanation of the made adjustments was added in lines 188-191:

“To compensate for the lack of SMD parts, the PCB thermal properties have been adjusted according to the electrical design of PCB. Two main materials of PCB have been taken under consideration - PCB laminate and copper. Given PCB design consisted volumetrically of around 0.6% of copper and 99.4% of PCB laminate.”

C2: Equation 3 is not clear, please re-write.

A2: This equation was given by the manufacturer of the sensor module and taken by the manufacturer of the thermistor. Therefore, the equation to be used for temperature calculation needs to be adjusted to the used thermistor.

C3: According to the authors, Figure 8 describes the whole software workflow/diagram of the software, however, the same figure refers to the data acquisition algorithm. Authors should clarify the difference. 

A3: Figure 8 was changed to be clearer, as well as its caption.

C4: On lines 159-150, authors claim that “The readability of the NFC interface is a requirement that has to be taken into account in the design process. The maximum layer of thickness was determined experimentally in previous work.” No reference to this work!

A4: The results of the named work were added in line 164 due to missing publications for this determination.

Yours sincerely

Lukas Boehler,

Reviewer 3 Report

The manuscript “Sensors in the autoclave - modelling and implementation of IoT steam sterilization procedure counter” by Lukas Boehler, Mateusz Daniol, Ryszard Sroka, Dominik Osinski and Anton Keller is devoted to studying the heat transfer from the outer surface of insulator to a temperature sensor by simulation methods.

Besides, the authors compared their simulation results with those obtained experimentally and found an acceptable concordance between them.

The manuscript is clearly written and may be of interest to readers of the journal involved in a development of sensors intended to work in a harsh environment. I have found a couple of typos such as the double repeat of thermal conductivity values at lines 211 (1.45) and 224 (0.14). Aside from these typos, the manuscript can be published as it is.

Author Response

Dear Sir or Madam,

First of all, we would like to express our gratitude to you for your explicit remarks

and constructive suggestions. We have carefully read your comments and made some modification on the original manuscript according to your comments and suggestions.

We hope to be able to meet your expectations. We first quote your comment, then give out our answer.

C1: I have found a couple of typos such as the double repeat of thermal conductivity values at lines 211 (1.45) and 224 (0.14).

A1: The document was checked, and mistakes were removed.

Yours sincerely

Lukas Boehler,

Round 2

Reviewer 1 Report

The authors taken into account my previous comments. However, as the discussion section is very small. I suggest to rename it to "discussion and conclusions", and the authors can merge the two sections. 

Author Response

Dear Sir or Madam,

thank you very much for your further suggestion for improvement. We merged the two chapters like you suggested.

Yours sincerely

Lukas Boehler,
